# Particle number-based trophic transfer of gold nanomaterials in an aquatic food chain

Fazel Abdolahpur Monikh [1,2 ✉], Latifeh Chupani[3], Daniel Arenas-Lago[4], Zhiling Guo[5], Peng Zhang [5], Gopala Krishna Darbha[6], Eugenia Valsami-Jones [5], Iseult Lynch [5], Martina G. Vijver [1], Peter M. van Bodegom [1] & Willie J.G.M. Peijnenburg [1,7]

Analytical limitations considerably hinder our understanding of the impacts of the physico-chemical properties of nanomaterials (NMs) on their biological fate in organisms. Here, using a fit-for-purpose analytical workflow, including dosing and emerging analytical techniques, NMs present in organisms are characterized and quantified across an aquatic food chain. The size and shape of gold (Au)-NMs are shown to control the number of Au-NMs attached to algae that were exposed to an equal initial concentration of $2.9 \times 10^{11}$ particles mL$^{-1}$. The Au-NMs undergo size/shape-dependent dissolution and agglomeration in the gut of the daphnids, which determines the size distribution of the NMs accumulated in fish. The biodistribution of NMs in fish tissues (intestine, liver, gills, and brain) also depends on NM size and shape, although the highest particle numbers per unit of mass are almost always present in the fish brain. The findings emphasize the importance of physicochemical properties of metallic NMs in their biotransformations and tropic transfers.

[1] Institute of Environmental Sciences (CML), Leiden University, Leiden, The Netherlands. [2] Department of Environmental and Biological Sciences, University of Eastern Finland, Joensuu, Finland. [3] South Bohemian Research Center of Aquaculture and Biodiversity of Hydrocenoses, Faculty of Fisheries and Protection of Waters, University of South Bohemia in Ceske Budejovice, Vodňany, Czech Republic. [4] Department of Plant Biology and Soil Science, University of Vigo, As Lagoas, Ourense, Spain. [5] School of Geography, Earth and Environmental Sciences, University of Birmingham, Edgbaston, Birmingham, UK. [6] Environmental Nanoscience Laboratory, Department of Earth Sciences and Centre for Climate and Environmental Studies, Indian Institute of Science Education and Research Kolkata, Mohanpur, West Bengal, India. [7] National Institute of Public Health and the Environment (RIVM), Center for Safety of Substances and Products, De Bilt, Bilthoven, The Netherlands. ✉email: f.a.monikh@cml.leidenuniv.nl

The adverse effects and fate of nanomaterials (NMs) in the environment must be properly understood to support the assessment of environmental risks of NMs. It was suggested that environmental risk assessment of quickly dissolving NMs may be achieved by read-across of the toxicity of the NMs to the corresponding soluble chemicals[1]. Some NMs that are persistent in the environment, however, raise additional concerns as they may, for instance, enter food webs and transfer through different trophic levels in food chains[2–4]. Understanding the trophic transfer of NMs is a challenging task, in part because numerous NM physicochemical properties (e.g., type, size, shape, and composition) may influence their trophic transfer. Moreover, NMs agglomerate and/or transform in the environment and organisms over time, confounding measurement[5,6].

Transfer of NMs from the environment to an organism may induce biotransformation and agglomeration of the NMs within the organism's body[7–9]. For example, agglomeration and/or dissolution of NMs were reported in the gut of some organisms[10–12]. Following uptake, interactions with proteins and metabolites alter the NMs, accelerating the dissolution of some metallic NMs, while corona formation may form a barricade slowing the dissolution of other types of metallic NMs[13,14]. Not only do these physical and chemical transformations change the properties of the NMs, which necessitates dynamic monitoring of the NMs, but they may also influence the gut uptake, biodistribution, and bioaccumulation of the NMs within the organisms, and thus NMs bioavailability to predators. Altogether, this combination of biological and physicochemical processes complicates generic predictions of the environmental risks of NMs, forcing each newly designed NM to be tested separately[15]. With the steep rise in the number of NM applications, this represents a major challenge in terms of time, cost, and resources, as well as raising ethical issues at the higher trophic levels.

To date, there are many methodological challenges in determining the uptake, transformation, and, finally, localization of NMs in organisms' bodies[16]. Existing ecotoxicity guidelines have been developed primarily for ecotoxicity testing of molecular chemicals, where the total mass of internalized potential toxicants is measured after the chemical digestion of the samples. Application of existing ecotoxicology guidelines yields information about the total mass of the element of interest in the tissue, but no conclusions can be drawn on the physicochemical properties of the internalized NM[17–19], such as size, shape, and number or whether the NM is still particulate. Due to their nanoscale properties, trophic transfer of NMs may not follow the patterns commonly seen for conventional contaminants, which are broadly predictable based on partition coefficients[20,21]. Particle dissolution, re-precipitation, and agglomeration following uptake influence NM size distributions and shapes in organisms across the food chain, but these transformations may not affect the total internalized mass of material (at a fixed time point). Reporting internalized dose as mass, as is commonly done[20,22], does not properly express the environmental risks of metallic NMs, as it gives no insight into the form in which the metal constituent exists[20,22,23].

In this study, we use particle number and mass as dose metrics to provide a comprehensive understanding of the trophic transfer of gold (Au)-NMs by monitoring the number and size distribution of the NMs in organisms along an assembled aquatic food chain consisting of microalgae, daphnids and fish. The dissolution and agglomeration of the particles in organisms is determined using a fit-for-purpose analytical workflow (in-house validated method) benchmarked against the standard method of mass-only detection. This allows us to determine how the initial shape and size of the Au-NMs influence their dissolution and agglomeration in each organism and how the organisms influence the NM size and shape following interaction/internalization, and how these processes influence the bioavailability of the NMs to the next trophic levels. Finally, we determine the trophic transfer of the NMs as a function of particle size and shape, and compare mass-based and number-based biomagnification factors (NBMFs). To tackle the analytical challenges in quantifying and characterizing NMs in organisms, particle extraction methods is developed and combined with modern characterization techniques, including single-particle (sp) and single-cell inductively coupled plasma mass spectrometry (ICP-MS). The scICP-MS allows quantitative analysis of the total mass of elements within individual cells with sensitivity at levels as low as attograms (ag) per cell (for algae)[20]. As a particle counting technique, spICP-MS allows measurement of the particle number concentration of metal-based NMs and differentiation between particulate and ionic forms of the elements[24,25].

## Results and discussion

**Characteristics of the NMs.** Commercially available spherical (10, 60, and 100 nm) and rod-shaped ($10 \times 45$ nm and $50 \times 100$ nm) citrate-coated Au-NMs from Nanopartz (USA) were characterized in Milli-Q (MQ) water in terms of particle size and morphology using transmission electron microscopy (TEM) (Supplementary Fig. 1). The physicochemical properties of the Au-NMs in MQ water are summarized in Supplementary Table 1. A negative zeta potential (a measure of colloidal dispersion electrostatic stability) was observed for all Au-NMs and ranged from −21 to −25 mV in MQ water and from −17 to −19 mV in the algal exposure medium (without algae). The stability of the particles against dissolution and agglomeration in the algal exposure medium without algae was monitored throughout the exposure duration (72 h). The dissolved fraction of the Au-NMs was <0.2% of the total Au mass for all particles (Supplementary Fig. 2a), which confirms the persistence of the NMs over the exposure duration of 72 h in the algal exposure medium. There were no significant changes in the measured mode of the size distribution and particle number of the Au-NMs in the algal exposure medium over 72 h of incubation (Supplementary Fig. 2b–f). This indicates that the particles are stable against agglomeration, which is likely due to the negative zeta potential increasing repulsion between the particles, thus preventing agglomeration.

**Association of Au-NMs with algal cells.** As primary producers, algae act as a gateway for the transfer of contaminants into aquatic food chains[26]. To understand how algae transfer NMs into aquatic food webs, *Pseudokirchinella subcapitata* was exposed to Au-NMs of different sizes and shapes. The initial concentration of the Au-NMs, to which the algae were exposed, was similar ($2.9 \times 10^{11}$ particles mL$^{-1}$) for all Au-NM shapes and sizes (Supplementary Table 2). No toxicity due to the exposure to Au-NMs was observed in the algae, as tested by measuring the chlorophyll content relative to unexposed controls (Supplementary Fig. 3), consistent with a previous publication[27]. After exposure, the unbound particles and the particles loosely attached to the algae were removed using a washing step with phosphate buffer saline (PBS, 0.1 M pH 7.5) followed by centrifugation (Supplementary Fig. 4) using Thermo Scientific Sorvall ST 16R Centrifuge. No dissolution of the Au-NMs was observed in PBS (from spICP-MS measurements that measure particles and ions) in agreement with our previous results[27]. We quantified the total mass of Au-NMs associated with the algal cells (the algae pellet) based on a cell-by-cell measurement using scICP-MS[28].

The data showed that the spherical 10 nm Au-NMs were found in 68% of the cells in the algal population, ranging from 20 to

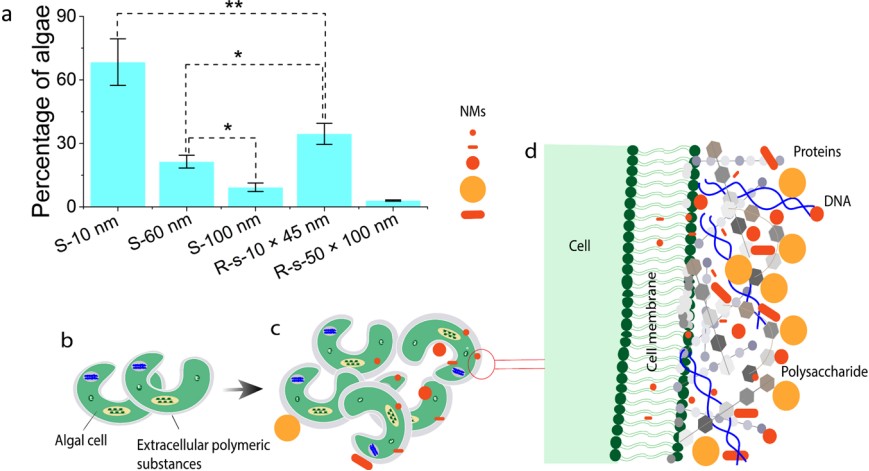

**Fig. 1 Association of Au-nanomaterials (Au-NMs) with algal cells. a** Percentage of algal cells that have associated Au-NMs as a function of particle size and shape. NMs, nanomaterials; S, spherical; R-s, rod-shaped. Error bars = SD ($n = 15$), biologically independent samples. Data were analysed using one-way ANOVA with Duncan's post hoc test (**$P < 0.001$; *$P < 0.05$). **b** *Pseudokirchinella subcapitata* algae with a thin layer of extracellular polymeric substances (EPS) around each cell in the natural condition. **c** The algal cells are exposed to Au-NMs of different sizes and shapes. The thickness of the EPS layer is expected to increase due to the exposure to Au-NMs as reported for other stressors[64]. The association of the Au-NMs is randomly distributed between the cells of the given population. **d** Penetration of Au-NMs of different sizes and shapes in the algal cell membrane and the EPS layer on the surface of the cells containing different biomolecules (e.g., polysaccharide, DNA, and proteins) excreted from the cell. The schematic shows how EPS may control the penetration of Au-NMs as a function of particle size and shape, where the smaller particles can penetrate, whereas the bigger or longer particles are trapped in the EPS. It is noteworthy that different elements in the schematic are not drawn to scale.

1200 ag Au per cell, whereas the rod-shaped 10 × 45 nm Au-NMs occurred in just 34% of the cells ranging from 30 to 200 ag Au per cell (Fig. 1a). As the first consumers (i.e., daphnids) feed on total algae, it is not necessary to differentiate between the NMs internalized in algal cells and those associated with the cells. The data showed that not all the cells within a given population accumulate Au-NMs, and that the Au-NMs were randomly distributed between the cells[20]. The surface of algae is covered by biopolymeric material (Fig. 1b) such as proteins, DNA, and polysaccharides, etc., known as extracellular polymeric substances (EPS). When algal cells are exposed to a stressor, the excretion of EPS by the cells increases (Fig. 1c) to protect the cells against the stressor[29]. It is likely that these substances determine the extent to which NMs associate with algae[20]. Smaller NMs may penetrate the EPS, while NMs with larger particle sizes/elongated shapes cannot penetrate this layer (Fig. 1d). The latter was then removed from the surface of the cells during the washing process.

**Trophic transfer of the Au-NMs.** Before describing the behaviour of the Au-NMs at each trophic level, we provide an overview of the percentage of the Au-NMs transferred to each trophic level. Trophic transfer of Au-NMs on a mass basis has been reported previously in different food chains[30,31]. Herein, we document the trophic transfer of Au-NMs as a function of particle physicochemical properties in an assembled aquatic food chain. The percentages were obtained from the calculated mass balance for the total mass of Au at different trophic levels (Supplementary Table 4) determined by ICP-MS. Accordingly, the total mass concentration of the Au-NMs at each trophic level was normalized to the total mass concentration of the Au-NMs at the previous trophic level. After exposure, the algae removed between 31% (rod-shaped 50 × 100 nm Au-NMs) and 90% (rod-shaped 10 × 45 nm Au-NMs) of the Au-NMs from the exposure media as a function of NM size and shape (Table 1). Nevertheless, only a small percentage of the Au-NMs were strongly associated with the algae, ranging from 0.01% for spherical 100 nm to 0.21% for the spherical 10 nm Au-NMs (Table 1). A considerable percentage (ranging from 31% to 89%) of the Au-NMs associated

with algae were removed from the cells after the washing process using PBS, indicating that NMs are mostly loosely attached or unbound to algal cells.

The total mass of the accumulated Au in daphnids fed with Au-NM-exposed algae (0.1 mg w.w. algae per daphnia) was between 0.73% (for spherical 60 nm Au-NMs) and 1.71% (for rod-shaped 10 × 45 nm Au-NMs) of the total Au in algae. The total mass of Au depurated from the daphnids was between 78% (spherical 60 nm Au-NMs) and 92% (rod-shaped 10 × 45) of the total mass of Au in 0.1 mg algae (Table 1), as shown by the depuration experiments in which the daphnids were treated with NMs-exposed algae for 72 h followed by 72 h of depuration. Although we could not clearly distinguish between the ionic and particulate form of the Au in the depuration medium, we could prove that Au, whether in the form of ions or particles, was excreted by the daphnids. It was previously documented that Au-NMs are depurated from daphnids, but no clear trend with the size or stabilizing agent could be found to explain the observed depuration rates[32]. This uptake and depuration data show that daphnids do not accumulate all NMs to which they are exposed to.

The fish (zebrafish) were fed with 10 daphnids (~100 mg of daphnids) each per day for 21 days. The results showed that a small percentage of the Au-NMs accumulated in daphnids transferred to fish (ranging from 0.03% for spherical 10 nm to 0.48% for rod-shaped 50 × 100) (Table 1). The total mass concentration of Au in the fish medium was measured on day 23 (48 h after the final feeding, whilst the medium had not been refreshed for 4 days). The results confirmed that a considerable fraction of the Au was excreted from the fish into the medium (ranging from 49% to 58% of the exposure dose) (Table 1).

**Transformation and agglomeration of Au-NMs in organisms.** Metallic NMs may dissolve within the organisms despite their stability in the exposure media[12,33–36]. For example, Briffa et al.[12] showed that the dissolution of cerium dioxides ($CeO_2$)-NMs in the exposure medium of daphnids is negligible, whereas, at simulated gut conditions of daphnids, there is around 40% Ce

**Table 1 Quantification of Au trophic transfer.**

| Au-NMs | Mass of Au removed by algae from the exposure medium/mass of Au in the exposure medium [%] | Mass of Au washed from the algae using PBS/mass of Au in the exposure medium [%] | Mass of Au accumulated in algae/mass of Au in the exposure medium [%] | Mass of Au in daphnia/mass of Au in 0.1 mg algae per daphnia [%] | Mass of Au depurated from daphnia/mass of Au in 0.1 mg algae per daphnia [%] | Mass of Au accumulated in fish/mass of Au in 100 mg daphnia per fish [%] | Mass of Au depurated form fish/mass of Au in 100 mg daphnia per fish [%] |
|---|---|---|---|---|---|---|---|
| Spherical 10 nm | 56 ± 8 | 76 ± 6a | 76 ± 12ab | 0.21 ± 0.01a | 1.09 ± 0.11 | 80 ± 14 | 0.03 ± 0.001c |
| Spherical 60 nm | 49 ± 11 | 46 ± 12cd | 45 ± 8d | 0.08 ± 0.004d | 0.73 ± 0.05 | 78 ± 9 | 0.14 ± 0.007d |
| Spherical 100 nm | 57 ± 8 | 62 ± 13ad | 62 ± 16ad | 0.01 ± 0.001c | 1.41 ± 0.11 | 84 ± 12 | 0.38 ± 0.14c |
| Rod-shaped 10 × 45 | 58 ± 12 | 90 ± 26ab | 89 ± 13b | 0.10 ± 0.03d | 1.71 ± 0.29 | 92 ± 14 | 0.04 ± 0.008c |
| Rod-shaped 50 × 100 | 51 ± 7 | 31 ± 10c | 31 ± 6c | 0.01 ± 0.002c | 1.38 ± 0.33 | 87 ± 15 | 0.48 ± 0.02b |

Percentage of the total mass of Au in each trophic level, determined by ICP-MS, normalized to the total mass of Au in the previous level, in the assembled aquatic food chain.
The calculated limit of quantification of the ICP-MS was 10 ng/L. The letters (a, b, c, d) show the significant differences between the NMs ($p < 0.05$).

dissolution from the particles. Nevertheless, in situ investigations on the biotransformation of NMs in organisms are challenging. Techniques such as synchrotron-based X-ray microanalysis allow in situ analysis to provide information about the presence of metallic NMs within tissues and offer information about the dissolution and speciation of the NMs and the released ions[34,37]. However, the typical resolution of synchrotron-based X-ray microanalysis (ca. 2–12 μm) currently is not sufficient to detect single NMs, particularly in organisms/tissue where the concentration of the NMs is expected to be low[38]. To overcome these limitations, the workflow developed in this study was tailored to the measurement of NMs and released ions at trace levels in situ in the tissue.

No detectable Au ion release was seen in the algae as measured by spICP-MS. Nonetheless, the NMs partially dissolved within the daphnids that had been fed with Au-NM treated algae. The percentages of intact particles and Au ions released from the Au-NMs in the daphnids are presented in Fig. 2a. The dissolution in the daphnids gut was NM size- and shape-dependent. The highest Au ion release in daphnids was detected for rod-shaped 50 × 100 nm Au-NMs, followed by spherical 100 nm NMs, which resulted in releases of 56% and 39% of the NM mass, respectively. The dissolution of the Au-NMs in the daphnids could have occurred in the gut, which has a lower pH (~4) than the exposure medium[12,39]. Absorption of a biomolecule corona to the surface of NMs when they enter the daphnids can also accelerate the dissolution of the particles if there is a strongly Au-binding protein present or can slow dissolution by promoting sulfidation processes[40]. The dissolution profiles in daphnids violated the common belief that smaller metallic NMs release more ions[41], despite the fact that the dissolution rate of smaller particles is anticipated to be more extensive than that of larger particles[42] due to a higher volume-specific surface area for ion release[20]. Apparently, Au-NMs do not follow this principle when they are accumulated in organisms. Possibly, the smaller Au-NMs agglomerate faster than the larger particles in the daphnids, with agglomeration reducing the particle dissolution due to decreasing the effective surface area of the particles, as reported previously for metallic NMs[43,44]. It is also possible that re-precipitation following dissolution occurred, whereby the smaller Au-NMs might dissolve faster than the larger ones, leading to higher local concentrations of Au ions and earlier onset of re-precipitation of Au ions onto particles. This biotransformation of metallic NMs in organisms is of paramount importance for understanding the toxicity of NMs as it determines the behaviour and biological fate of the NMs and the release of ions from the particles and, subsequently, the target organs reached by the different metal forms[14].

To test this hypothesis, we comprehensively quantified the number-based size distribution of the accumulated particles at each trophic level (Supplementary Fig. 5a, b) after extraction from the organisms using a method that had been shown previously not to alter the NMs size distribution (see "Methods" section). The shift in mode ($M_{size}$) of the size distribution between two subsequent trophic levels shows how the size distribution of Au-NMs is affected across the food chain (Table 2 and Supplementary Fig. 5a, b). The $M_{size}$ of the spherical 10 nm and the rod-shaped 10 × 45 nm Au-NMs in daphnids shifted towards a larger particle size (from 20 to 32 nm and from 25 to 28 nm, respectively). The $M_{size}$ of spherical 60 nm, spherical 100 nm and rod-shaped 50 × 100 nm Au-NMs in daphnids shifted towards smaller particle size compared to Au-NMs in algae, confirming our findings that daphnids modify the size distribution of these Au-NMs by dissolution (Fig. 2a). The $M_{size}$ of the transformed spherical 10 nm and the rod-shaped 10 × 45 nm NMs in daphnids was almost equal to the $M_{size}$ of the 60 nm and 100 nm spherical Au-NMs and rod-shaped 50 × 100 nm Au-NMs (Table 2), ranging from ~25 to 40 nm mass-based size, suggesting that in fact, the hypothesis related to faster dissolution and re-precipitation of the smaller particles is the most likely explanation for the transformation process. Thus, irrespective of the initial size, all Au-NMs were transformed into roughly equivalent sizes in the daphnids. By modifying the size of the particles, daphnids can dramatically influence the biological fate of NMs in the higher trophic levels and how metallic NMs distribute over the entire aquatic food chain.

No Au ions were detected in fish tissues, indicating that (a) the Au-NMs in fish did not dissolve, and (b) the ions released from the particles in the daphnids were not detectable in fish. Two explanations can be put forward for these findings. First, the Au

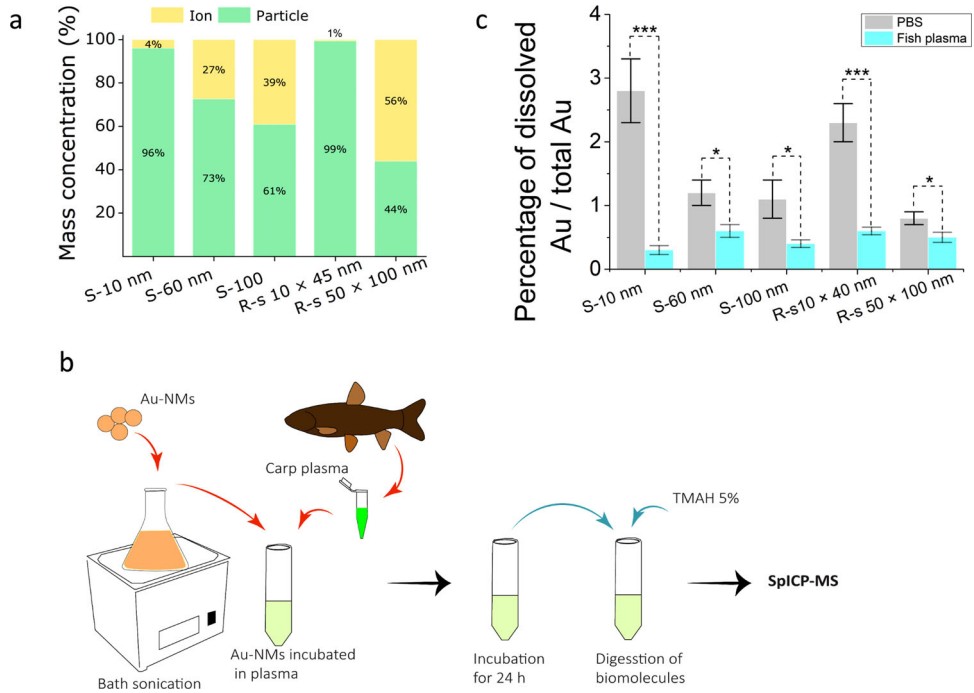

**Fig. 2 Concentrations of gold nanomaterials (Au-NMs) and released Au⁺ from the Au-NM. a** Percentage of the intact particles (green) and Au ions (yellow) released from the Au-NMs in daphnids ($n = 5$). The percentage was calculated by normalizing the mass of the particles or the mass of Au ions in daphnids to the total mass of Au in the daphnids. The graph shows that a higher percentage of dissolution in the daphnids occurred for the rod-shaped 50 × 100 nm followed by spherical 100 nm Au-NMs. S: spherical, R-s: rod-shaped. **b** Illustration of the incubation of Au-NMs in fish plasma extracted from adult carp. The Au-NMs were sonicated for 5 min and incubated in 10 times diluted fish plasma for 72 h. After incubation, the sample was digested with 5% Tetramethylammonium hydroxide (TMAH) and the concentration of the released ions was measured using single-particle inductively coupled mass spectrometry (spICP-MS). **c** Concentration of Au ions released from 10 mg mL$^{-1}$ Au-NM of different sizes and shapes after incubation in fish plasma for 72 h. One-way ANOVA with Duncan's post hoc test were used to analyse the data (***$P < 0.0001$; *$P < 0.05$). Error bars = SD ($n = 15$), independent samples.

**Table 2 The mode of the size distribution measured for Au-NMs in different organisms and in fish tissues.**

| NMs | Mode (nm) | | | | | |
| --- | --- | --- | --- | --- | --- | --- |
| | Algae | Daphnia | Fish | | | |
| | | | Intestine | Liver | Brain | Gills |
| Spherical 10 nm | 20 ± 5 | 32 ± 3 | 42 ± 4 | 44 ± 6 | 50 ± 6 | 47 ± 6 |
| Spherical 60 nm | 60 ± 3 | 33 ± 5 | 38 ± 6 | 41 ± 4 | 61 ± 7 | 86 ± 12 |
| Spherical 100 nm | 97 ± 8 | 38 ± 5 | 45 ± 7 | 42 ± 3 | 58 ± 9 | 39 ± 5 |
| Rod-shaped 10 × 45 nm | 25 ± 3 | 28 ± 6 | 47 ± 3 | 93 ± 13 | 58 ± 8 | 43 ± 10 |
| Rod-shaped 50 × 100 nm | 61 ± 8 | 36 ± 6 | 44 ± 11 | 61 ± 8 | 50 ± 7 | ND |

ions in daphnids may not be bioavailable to fish and consequently, they were not taken up by fish. Many studies have already shown that the distribution of soluble metals in organisms influences their trophic transfer by controlling their bioavailability for predators[45,46]. For example, soluble metals distributed in subcellular organelles are more efficiently taken up by predators than are metals present in insoluble fractions within prey, such as in metal-rich granules[45]. It is likely that the ionic form of the Au in daphnids was stored in fractions that make them non-bioavailable to fish. Second, the absence of Au ions released from the particles in fish could be due to the evolution of the biomolecule corona on the surface of the particles when entering the fish body[47–49]. The corona may reduce the Au release from Au-NMs in fish unlike in daphnids, as observed for other NMs in other organisms[50–52]. We tested this hypothesis by incubating 10 mg L$^{-1}$ of the Au-NMs in a 1 mL fish plasma (10

times diluted with PBS; containing 4.5 g L$^{-1}$ proteins) for 24 h (Fig. 2b and Supplementary Section 8). After incubation, the samples were digested using Tetramethylammonium hydroxide (TMAH) to remove the biomolecules and free the Au ions potentially sorbed to these molecules (for detailed information about the method see Supplementary Section 8). The residuals were measured using spICP-MS to differentiate between Au ions and Au-NMs. The results (Fig. 2c) showed that the biomolecule corona of fish plasma indeed reduced the dissolution of all the Au-NMs, regardless of particle size and shape, when compared to the dissolution of the NMs in ultrapure water.

Although there was a slight increase in the $M_{size}$ of the Au-NMs, the size distribution of the Au-NMs in the intestine of fish remained almost the same as that in daphnids. The increase in the $M_{size}$ of spherical 10 nm (increased from 32 to 42 nm) and rod-shaped 10 × 45 nm (increased from 26 nm to 47 nm) Au-NMs

was more pronounced than the increase in the other Au-NMs. These results suggest that the spherical 60 nm, the spherical 100 nm and the rod-shaped $50 \times 100$ nm Au-NMs did not dissolve further or agglomerate in the fish gut. It is not surprising that the spherical 10 nm and rod-shaped $10 \times 45$ nm continue agglomerating even when their larger counterparts are stable against agglomeration, because they have a significantly larger volume-specific surface area compared to the larger counterparts[20], which enhances particles agglomeration. This finding indicates that Au-NMs may not undergo biotransformation in fish, suggesting that the transformation of metallic NMs in organisms is a species-dependent phenomenon. This even questions the application of model organisms to assess the risk of NMs as each organism may have different influences on the biotransformation of NMs and the findings cannot be extrapolated to other organisms.

The results furthermore indicated that Au-NMs were selectively distributed across the fish tissues because a different size distribution of Au-NMs was measured in various tissues (Table 2 and Fig. S5a, b, Supplementary Information). Particles with $M_{size}$ between 50 and 61 nm accumulated in the brain regardless of their size in the daphnids and fish intestine. Because fish were exposed to Au-NMs only through food, we did not expect to measure detectable numbers of Au-NMs in the gills. Nevertheless, particles with $M_{size}$ between 39 nm (spherical 100 nm Au-NMs) and 86 nm (spherical 60 nm Au-NMs) were detected in the gills. There are two possible explanations for this: first, the particles may be transferred via the blood to be excreted from the body through the gills[53] and the smaller size of the particles allowed them to penetrate the gills. Second, the particles may be excreted from the fish gut into the culture medium and be subsequently taken up by fish through the gills. The high percentage of excreted Au from fish (Table 1) suggests that most of the accumulated Au-NMs in the gills are likely taken up from the medium rather than translocated following ingestion.

**Number of accumulated Au-NMs at each trophic level**. To provide insight into the number-based trophic transfer of Au-NMs, we measured the number of Au-NMs at each trophic level (Algae–Daphnia–Fish) using spICP-MS. A schematic representation of the Au-NMs trophic transfer as a function of NM physicochemical properties is illustrated in Fig. 3a. Our results (Fig. 3b) showed lower numbers of Au-NMs in daphnids (ranging from $3 \times 10^4$ for rod-shaped $50 \times 100$ nm NMs to $6 \times 10^5$ for spherical 10 nm NMs) compared to algae and lower again in the fish (ranging from $1.5 \times 10^4$ for rod-shaped $50 \times 100$ nm Au-NMs to $2.5 \times 10^4$ for rod-shaped $10 \times 45$ nm Au-NMs) than in daphnids. Previous studies have shown that biofilms and algae could constitute a potential entry route for NMs into aquatic food chains[3,54]. Herein we show that not all particles attached to algae were transferred to higher trophic levels. However, unlike the algae, all the daphnids and fish measured contained NMs.

In this study, we demonstrate that although the algae were exposed to the same total number of NMs of the different sizes and shapes, the number of NMs associated with the algal cells was a function of NM size and shape (Fig. 3b), ranging from $4.6 \times 10^7$ for spherical 100 nm NMs to $5.6 \times 10^8$ for spherical 10 nm NMs. It was also detected that not all algal cells had a similar number of NMs associated with their surfaces and the NMs were heterogeneously distributed across the algal cell population, as reported previously[20,28]. The daphnids controlled the number of NMs transferring to fish and the biodistribution of the Au-NMs in fish by modifying the size distribution of the Au-NMs (Fig. 3b). Herein we show that trophic transfer of Au-NMs, as determined on the basis of particle number, depends on the physicochemical properties of the NMs and the species.

To understand the biomagnification of the Au-NMs as a function of particle size and shape, the NBMFs were calculated for each pair in the food chain separately (Algae–Daphnia, Daphnia–Fish) and for the entire food chain of Algae–Fish (Fig. 3c). The NBMF of the Algae–Fish was calculated by dividing the total number of the Au-NMs in fish by that in algae. The NBMF of all Au-NMs was higher in the Daphnia–Fish pair compared to Algae–Daphnia (Fig. 3d). We obtained different results for ionic Au (released from the Au-NMs), which showed no transfer from daphnids to fish. Interestingly, the NBMFs of spherical 10 nm and rod-shaped $10 \times 45$ nm were lower in the Daphnia–Fish and Algae–Fish compared to the other Au-NMs (Fig. 3d). This indicates that only a small portion of the spherical 10 nm and rod-shaped $10 \times 45$ nm transfer to fish compared to the other Au-NMs tested. This suggests that particle size plays a significant role in the trophic transfer of NMs and the bioavailability of small NMs to fish may be lower than the bioavailability of their larger counterparts.

All of the Au-NMs transferred from daphnids to zebrafish, except for the rod-shaped $10 \times 45$ nm Au-NMs, accumulate most profoundly in the brain, ranging from 45% for spherical 10 nm particles to 59% for rod-shaped $50 \times 100$ nm Au-NMs (Fig. 4a). Although the ability of Au-NMs to pass the blood–brain barrier and penetrate the fish brain has been documented previously[53,54], in this study the number of particles that can accumulate in the brain was documented as a function of their properties. It is possible that some of the NMs could be in the vascular compartment of the brain, i.e. the endothelial cells that comprise the blood–brain barrier itself, which has been demonstrated in vitro to be a site of NM accumulation[55]. As the workflow presented in this study has been developed using a set of robust techniques to distinguish between particle and ions, we were able to reproducibly measure the number of Au-NMs in the fish brain tissue (including the endothelial barrier). In general, understanding the number of NMs in the brain is important to determine the effective dose because NMs have a larger surface area than the corresponding bulk material, which means a higher number of molecules on the surface of NM are available for interaction with surrounding biological material.

To date, no study is available to show how the dietary uptake of Au-NMs influences their biodistribution in the fish's body. Our findings on the biodistribution deviate from those of medical-oriented studies on the distribution of Au-NMs in a variety of organisms. Upon direct injection in the body, as performed in medical-oriented studies, Au-NMs tend to accumulate mainly in the liver, spleen, and kidney[56,57]. However, it is likely that when Au-NMs are taken up via the diet, they are more susceptible to transfer to the brain. For example, Unrine et al.[31] reported that the bioavailability of NMs through trophic exposure is higher than through direct exposure and that trophic transfer may influence the way NMs biodistribute in the bodies of organisms. No general rule for NM accumulation can be established as yet though, due to differences in the research methods and dose metrics applied.

The calculated NBMF of Au-NMs from algae to fish are reported for each fish tissue in Fig. 4b. The mass-based biomagnification factors (MBMFs) are reported in Supplementary Fig. 6. The MBMF showed a similar trend as that determined for NBMF, confirming that there is no dissolution of the Au-NMs in fish, as the calculated mass is proportional to the total mass of the measured particles in fish. The highest number of particles ended up in the brain followed by the liver, as compared to other tissues, except for the rod-shaped $10 \times 45$ nm NMs. This is in line with suggestions that the liver and brain could be targets for NMs[58–61] and stresses the importance of evaluating the fate of NMs in fish brains to elucidate whether they undergo a

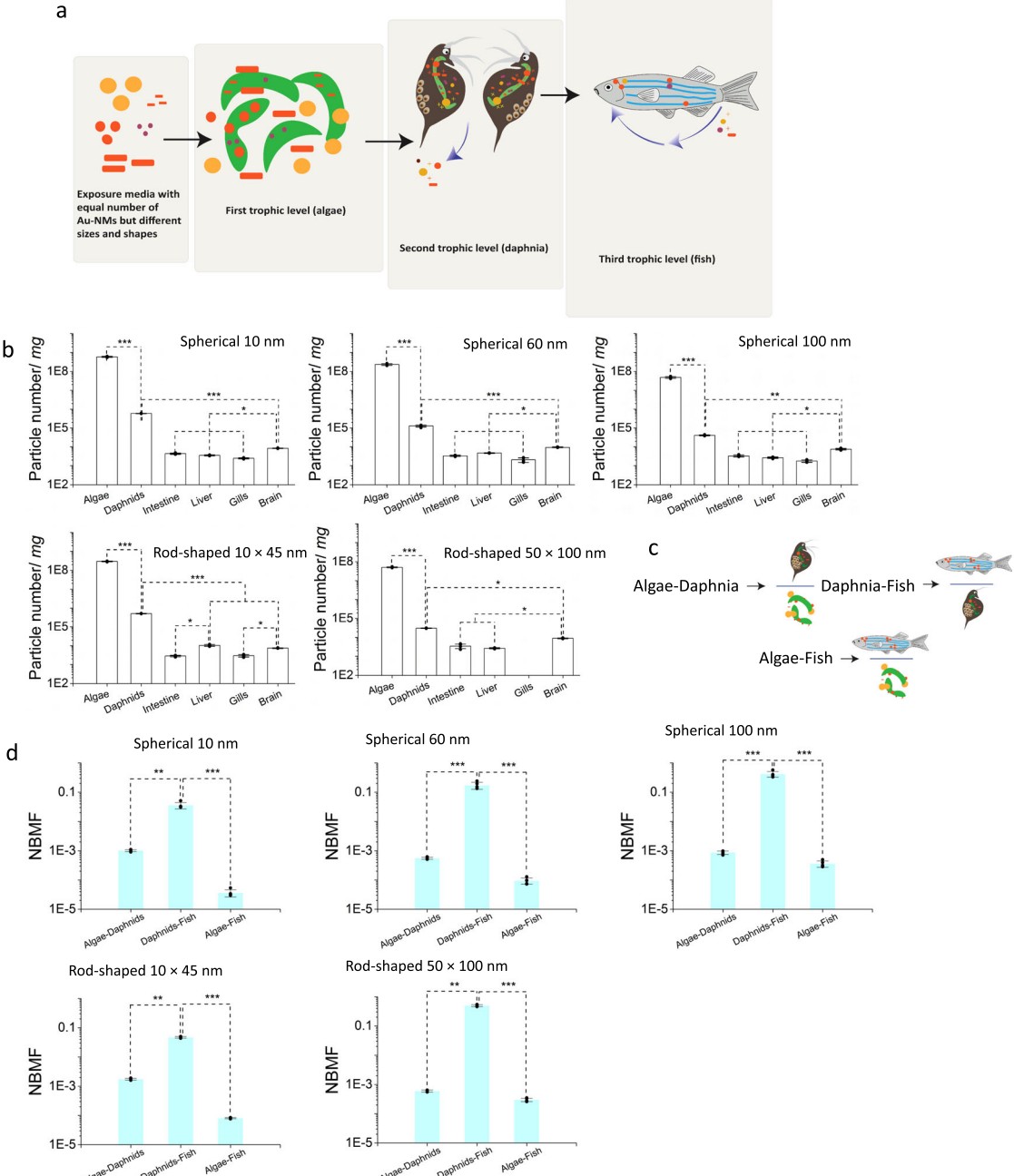

**Fig. 3 The number of Au-NMs at each trophic level. a** Schematic representation of the trophic transfer of NMs based on particle number. Despite being exposed to the same number of Au-NMs, algae accumulate (associated with the cell or internalized in the cells) different numbers of Au-NMs as a function of NM size and shape. Smaller numbers of the Au-NMs accumulate in the daphnids compared to the algae. The NMs undergo dissolution and aggregation in the daphnids gut as a function of the NM properties, which changes the size distribution of the NMs accumulated in the daphnids. A high percentage of the NMs are excreted from the daphnids and only a small fraction is transferred to the fish. No particle number-based biomagnification occurs in fish. However, the number of accumulated NMs in fish is a function of the initial NM size and shape. The biodistribution of the Au-NMs in fish was also NM size- and shape-dependent, where a higher number of particles were detected in the brain compared to other tissues. **b** Particle number (per mg wet weight tissue) of Au-NMs accumulated at each trophic level (algae, daphnids, and fish tissues) as a function of NM size and shape. **c** Schematic representation of the trophic transfers (Algae–Daphnia, Daphnia–Fish, and Algae–Fish) considered to calculate the number-based biomagnification factors (NBMF). **d** The calculated NBMFs of the Au-NM in the different trophic levels of the assembled food chain used in this study. Note that the trophic transfer of Algae–Fish was calculated by dividing the total number of the particles in fish by the total number of the particles in algae, although the fish was not fed with algae directly (**P < 0.001, ***P < 0.0001). S: spherical, R-s: rod-shaped. Error bars = SD, biologically independent samples (n = 5). Data were analysed using one-way ANOVA followed by Duncan's post hoc test (**b**, **d**) (***p < 0.0001; **p < 0.001; *p < 0.05).

transformation in the brain or accumulate in the particulate form. Nevertheless, overall there was no NBMF in fish, i.e., the numbers of particles in fish tissues were very low and not higher than those in algae. Our finding is in disagreement with previous studies showing the occurrence of biomagnification from biofilm to snail (biomagnification factor > 1.8)[62] and from bacteria to protozoa (biomagnification factor > 5)[4], which could be due to the differences in the food chains examined, including the feeding

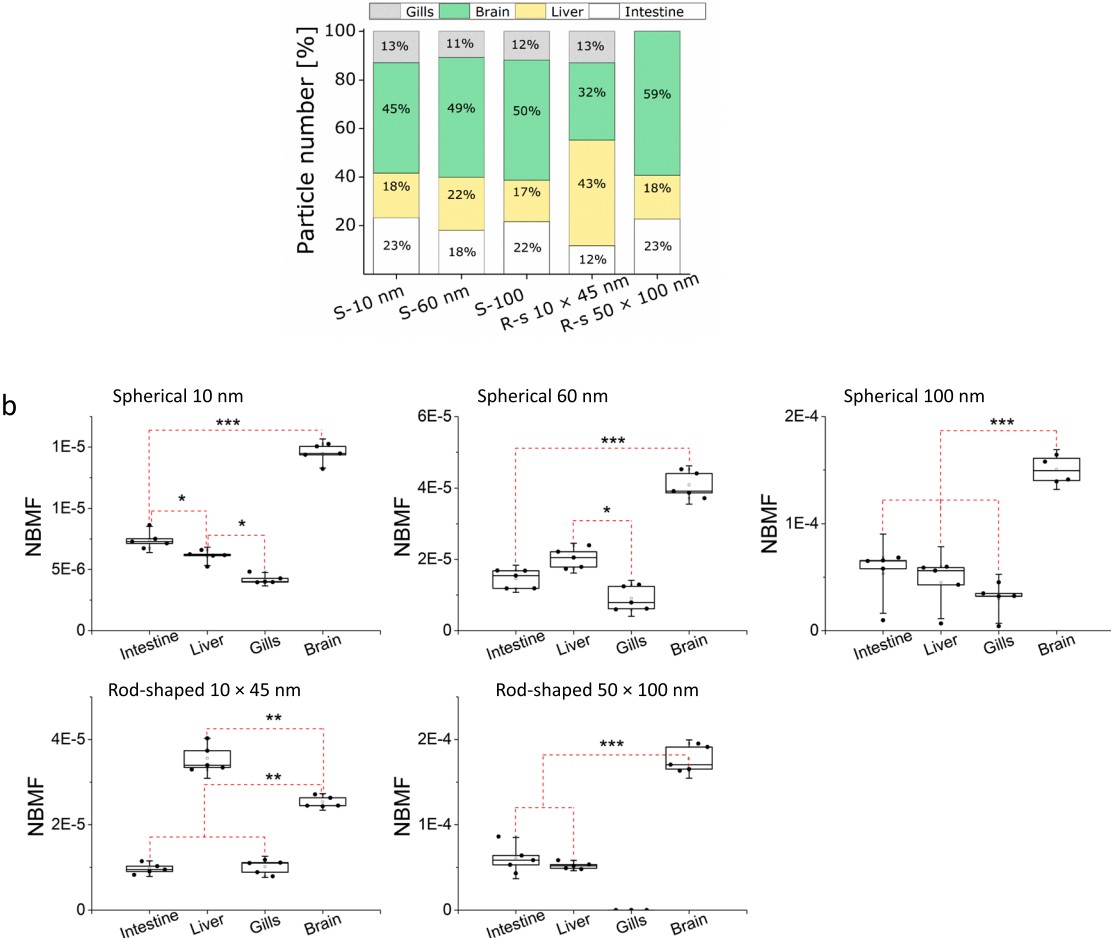

**Fig. 4 Number-based biomagnification factor (NBMF). a** Percentage of the total accumulated particle number in different tissues of fish ($n = 5$). The graph shows the number-based biodistribution of the Au-NMs in fish as a function of particle shape and size. **b** NBMFs of the Au-NM in fish obtained by dividing the concentration of the Au-NM in tissues of fish by the concentration of the Au-NMs in the algae. Data ($n = 5$) were analysed using one-way ANOVA followed by Duncan's post hoc test (**b**) (***$p < 0.0001$; **$p < 0.001$; *$p < 0.05$). Box plots (**b**) indicate median (middle line), 25th, 75th percentile (box), and SD (whiskers).

patterns of the organisms involved and whether or not the organisms transform the NMs. It is also related to the fact that the NMs underwent significant dissolution in the daphnia guts but the fish did not accumulate the dissolved Au, which may not be the case for other organisms.

Our results show that persistent NMs such as Au-NMs have the potential to transfer to higher trophic levels via an aquatic food chain. Although algae were exposed to similar numbers of each Au-NM size and shape (spherical 10 nm, spherical 60 nm, spherical 100 nm, rod-shaped 10 × 45 nm and rod-shaped 50 × 100 nm), the EPS provides effective removal of larger particles, leading to a higher initial accumulation of the smaller Au-NMs (spherical 10 nm and rod-shaped 10 × 45 nm) on algae. The association of NMs to algae, as an important gateway for NMs entering aquatic food webs, strongly depended on NM shape and size. Daphnids filter-feed on algae and ingest a considerable amount of algal-associated NMs (between 80%–90% of Au-NMs in algae). However, only a small fraction of the NMs accumulate in daphnids (between 0.73% and 1.7% of the NMs associated with the algae), indicating that Au-NMs are not potentially bioavailable in high concentrations to the higher trophic levels in the aquatic food chain. Daphnids modulates the size distribution of the Au-NMs, through the dissolution of the larger Au-NMs and dissolution-re-precipitation and agglomeration of the smaller

Au-NMs, leading all the NMs to have similar sizes (ranging from ~25 nm to 40 nm mass-based size) in daphnids. Only a small fraction of the Au-NMs (between 0.03%–0.48% of the Au-NMs in daphnids) transferred from daphnids to fish. No further transformation and agglomeration of the Au-NMs occurred in fish, but biodistribution was observed in fish, with the brain and liver as the target organs. Excretion of Au-NMs from fish led to re-uptake via fish gills. Our findings, therefore, illustrate what could happen after long-term exposure to Au-NM in a three levels aquatic food chain. Using NM number as a dose metric facilitates the comparison between results reported by different laboratories and provides important insights into the transformations undergone during the tropic transfers which would be lost upon applying mass as the only dose metric. Future research should use the developed analytical workflow to focus on assessing and quantifying other types of NMs and other food chains, given that we observed that the biotransformation and behaviour of NMs in organisms could be species-dependent.

## Methods

**Chemicals and materials.** All chemicals were reagent grade and were used without further purification unless noted otherwise. Water was deionized by reverse osmosis and purified by the Millipore MQ system. Au-NMs were purchased from

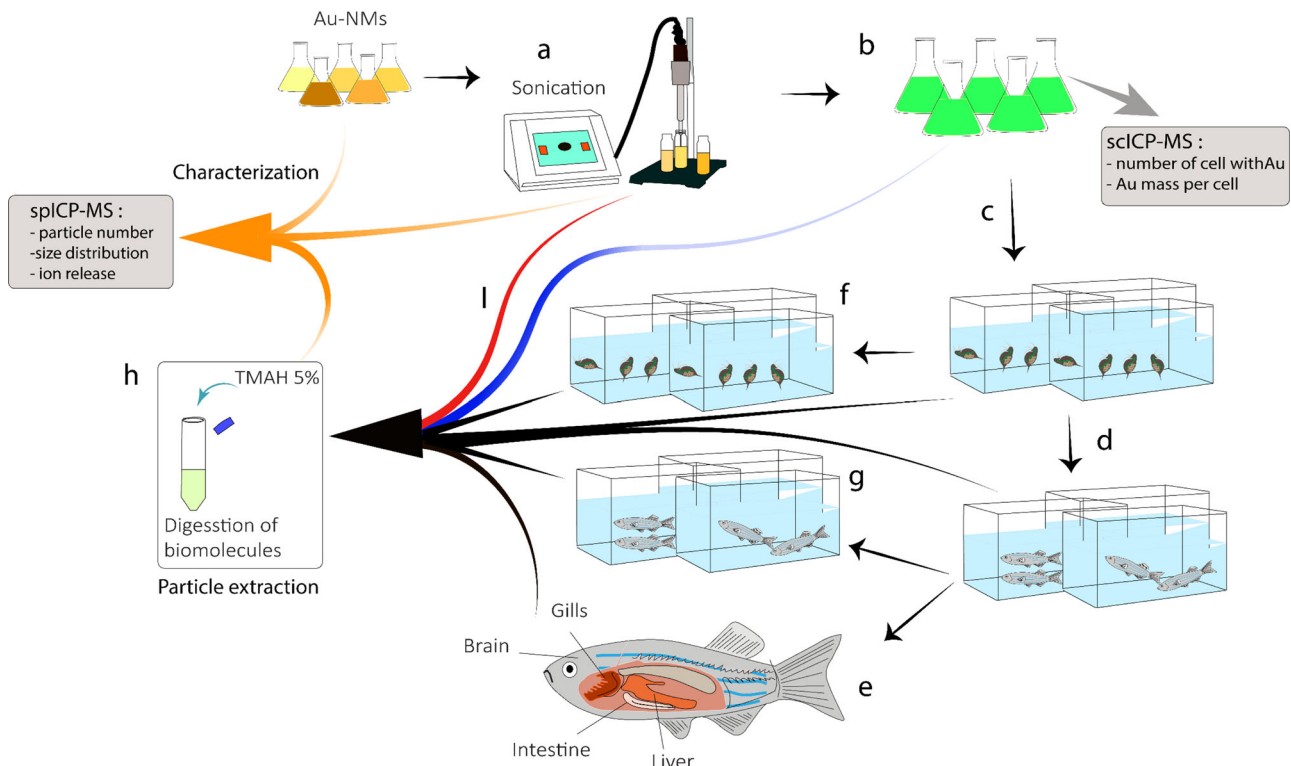

**Fig. 5 The experimental design and method validation.** The gold (Au)-nanomaterials (NMs) were dispersed in Milli-Q (MQ) water and sonicated using a tip sonicator (**a**). Some of the Au-NM dispersion samples were measured using single-particle inductively coupled plasma mass spectrometry (spICP-MS) before and after the sonication (green arrows) to evaluate the influence of the sample preparation process on the Au-NMs physicochemical properties. Some of the samples were used to check the influence of the NM extraction method (without organisms) on the physicochemical properties of the Au-NMs (red arrow). **b** After sonication, the Au-NMs were immediately used to expose the algae for 72 h. Aliquots of the algae were used to analyse the Au-NMs using spICP-MS (grey arrow) and to analyse the cellular association of the Au-NMs using single-cell (sc)ICP-MS (blue arrow). The rest of the Au-NM-exposed algae were used to feed the daphnids. **c** The daphnids were fed with the Au-NM-exposed algae for 72 h. Some of the exposed daphnids were used to analyse the Au-NMs accumulated in the organisms using spICP-MS (grey arrow) and some were used to perform the depuration experiment (**d**). After depuration, aliquots of the medium used for the depuration experiment were analysed using spICP-MS and ICP-MS (**d**, grey arrow). The reset of the daphnids was used to feed the fish. **e** Fish were fed with exposed daphnids for 21 days. Accordingly, daphnids were exposed to algae for 72 h on a continuous basis to always have freshly exposed daphnids available for feeding fish. Some of the treated fish were used to perform the depuration experiment (**f**) and some were dissected to remove the intestine, liver, gills, and brain (**g**). The depuration medium and the dissected tissues were analysed by spICP-MS. **h** Before analysing the samples using spICP-MS, the NMs were extracted from the samples using 5% Tetramethylammonium hydroxide (TMAH), which had previously been shown not to alter Au-NMs size distribution by treating particles in the absence of the tissue and monitoring any changes in the NMs size and size distribution (see below for details). After NMs extraction, the samples were analysed using spICP-MS (**h**, green arrow).

Nanopartz (Loveland, CO, USA) and used as received. The TMAH (25%) was purchased from Sigma–Aldrich.

**Characterization of Au-NMs in algal exposure medium.** The size and morphology of the pristine Au-NMs dispersed in MQ water were determined using TEM (JEM 1400). Zeta potential measurements were performed using a Zetasizer Nano device (Malvern Panalytical, Almelo, the Netherlands). The dissolution profiles of the particles in the algal exposure medium were detected using spICP-MS (PerkinElmer NexION 300D ICP-MS, see Supplementary Section 10 for more information). Characterization and quantification of the Au-NMs in the different organisms required method development and in-house method validation, as described in the next sections (Fig. 5).

**Microalgae exposure.** The schematic of the experimental design and algal exposure is illustrated in Fig. 5 and detailed information about culturing the algae is reported in Supplementary Section 11. Briefly, the unicellular algae *P. subcapitata* was cultured in algal medium Woods Hole and used as the first trophic level. The algae density used was $5 \times 10^3$ cells/mL in accordance with the OECD recommended guideline (OECD 201) for initial biomass of algae, and was measured using an AquaFluor fluorometer (Turner Designs, San Jose, CA, USA). The exposure was conducted immediately after sonicating the Au-NMs dispersion using a SONOPULS ultrasonic homogenizer (BANDELIN Electronic, Berlin, Germany) for 10 min at a delivered power of 40 W (Fig. 5a). Algae were exposed (Fig. 5b) to $2.93 \times 10^{11}$ particles mL$^{-1}$ in the algal culture medium containing one of the different particle sizes or shapes during their steady-state phase of growth (6–7 days). The flasks containing the exposed algae were placed in a climate

chamber (22 °C) at a light intensity of 70 mE m$^{-2}$ s$^{-1}$ for 72 h. The exposed algae were collected and the loosely attached and unbound Au-NMs were removed from the surface of the algae, as described in Supplementary Section 5. Aliquots of the algal samples were separated for scICP-MS (to quantify the number of cells) (Fig. 5b, blue arrow) and for spICP-MS (to quantify the number of Au-NMs in/on the cells) analysis (Fig. 5b, grey arrow).

**Algal sample preparation and measurement using scICP-MS.** After removing the particles that were unbound and loosely attached to the cells, the cells were prepared for measurement using scICP-MS. Accordingly, the suspensions obtained were centrifuged at $2000 \times g$ and 4 °C for 10 min (using Thermo Scientific Sorvall ST 16R Centrifuge) and the supernatants were separated. The pellets were diluted with 5 mL of PBS and the number of cells was measured using an Aquafluor metre. Approximately 2 mL of each algal sample was used for scICP-MS measurement. No visible Au-related signals were detected in the control samples, which demonstrates that no background Au or additional interferences influenced the signals. A PerkinElmer NexION 300D ICP-MS was used to perform scICP-MS (Supplementary Table 6). The sample introduction of the cells to the scICP-MS was described in our previous study[20]. Briefly, an Asperon spray chamber introduced a low volume of algal cell dispersion at a rate of 0.02 mL/min into the plasma without damaging the cell membrane. A high-efficiency quartz concentric nebuliser (MEINHARD HEN) was applied and the dwell time and acquisition time were set at 50 μs and 40 s, respectively, where each detected event (peak) represents a cell. Thus, cells which contain Au-NMs are measured on a cell-by-cell basis. From the total cell count with the Aquaflour metre, we determine the number of cells with no Au-NMs.

**Daphnids exposure**. Adult *Daphnia magna* were cultured in a culture room at 22 °C on a 12 h light–dark cycle. Acclimatization to the culture room's conditions was permitted for 1 week. According to the OECD test guideline 202, the fresh daphnids culture medium (Elendt M7) was prepared every day and aerated with an air pump. Medium pH was controlled at pH 8. Organisms were fed with a few drops of unexposed *P. subcapitata*. For exposure of the daphnids to NM-exposed algae (Fig. 5c), adult *D. magna* individuals were divided into 6 groups (90 daphnids per group). Each experiment was conducted in triplicate. Algae that were exposed as described above to the NMs were used to feed the daphnids. Control groups were fed with unexposed algae. Daphnids were fed with 0.1 mg of the NM-exposed algae at time points 0, 24, and 48 h of the exposure duration and cultured for 72 h. The total number of surviving individuals for different groups was calculated and dead daphnids were separated daily. After 72 h, the daphnids were harvested, washed three times with MQ water, and then immediately fed to zebrafish. Some of the daphnids were separated for the depuration experiment (Fig. 5d) and some for Au-NMs characterization using spICP-MS (Fig. 5c, grey arrow). Depuration experiments were performed to allow the organisms to excrete the fraction of the Au-NMs that could not pass the gut epithelium and internalize into the organisms[32,63]. For the depuration experiment, after 72 h of exposure (feeding with Au-NM-exposed algae), the daphnids were washed five times with pure water and then placed into a clean culture medium for 72 h without feeding. Aliquots of the culture medium were taken at 72 h to measure the total mass of Au excreted from the daphnids using ICP-MS (Fig. 5d, grey arrow).

**Zebrafish exposure and tissue dissection**. All experiments were performed with the approval of the ethics committee of the Research Center of Aquaculture and Biodiversity of Hydrocenoses, the University of South Bohemia in Ceske Budejovice, Czech Republic (MSMT-6744/2018-4). Adult zebrafish (*Danio rerio*) were maintained for 30 days in the laboratory conditions to become acclimated to the zebrafish culture water (carbon-filtered, dechlorinated tap water). The zebrafish were kept at 22 °C on a 14 : 10 h light : dark cycle. During acclimatization, zebrafish were fed with unexposed live *D. magna* (10 daphnids, which were equal to almost 100 mg w.w., per day). Exposures of zebrafish were conducted in 30 L glass aquaria and water was changed every 36 h. Each fish was fed for 21 days with 10 Au-NM-exposed daphnids per day (Fig. 5e). After exposure, the fish were held for 48 h without feeding (Fig. 5f) to empty their stomachs and the concertation of the total Au in the medium was measured to obtain the Au depuration by fish (Fig. 5f, grey arrow). The intestine, liver, gills, and brains of each fish were immediately dissected (Fig. 5g), weighed, and digested for NM extraction. The extracted NMs were prepared for spICP-MS analysis (Fig. 5g, grey arrow) to measure the number of the particles in each tissue, the particle sizes and the fraction (if any) of dissolved ions.

**Particle extraction method**. To extract Au-NMs from the organisms, aliquots of the algae, daphnids, and fish tissues from each treatment (Fig. 5, grey arrows) were separated and digested using 5% TMAH (see Supplementary Section 13 for more details). We evaluated the performance of the NM extraction method (Fig. 5, red arrow) and sample preparation (Fig. 5, green arrows), e.g., dispersion stability after sonication, organic materials (organisms tissues) removal and sample handling to ensure minimum influence on the Au-NMs stability, including dissolution, and agglomeration. The results (Supplementary Fig. 8) showed that the application of the TMAH and sample preparation approach did not influence the particle size distribution, confirming no influences on the particles. This in-house validated method was used for Au-NMs extraction from the organisms for further analysis. The particles extracted from the organisms were quantified using spICP-MS, which quantifies the number of Au-NMs and the particle sizes as well as the amount of dissolved Au ions. We must mention that the samples resulting from the depuration experiments were also measured using spICP-MS; however, no particles were detected, which could be due to the low concentration of the particles in the depuration media.

**Quantification of NMs particle number concentration using spICP-MS**. All spICP-MS measurements were performed on a PerkinElmer NexION 300D ICP-MS operating in sp mode. The operational parameters for spICP-MS are summarized in Supplementary Table 5. Dispersions of Au-NMs with sizes of 10, 60, and 100 nm and mass concentration of 50 mg/L were used to determine the transport efficiency[10]. Stock standards of Au (100 μg/L) were prepared using MQ water. Calibration standards in the concentration range 1–3 μg/L were prepared by diluting the corresponding ionic stock standards further in MQ water. Particle sizes, particle numbers, and mass concentrations were determined according to the method described in detail and validated in our previous work[10].

**Data analysis and calculation**. The graphs were plotted using OriginLab 9.1. Data were evaluated statistically for normality using a Kolmogorov–Smirnov test in SPSS version 23.0 following the method reported in a previous study[10]. One-way analysis of variance, followed by Duncan's post hoc test, was performed to determine statistically significant (two-sided) differences between samples. The NBMF was calculated for each trophic level (Eq. 1) and for each fish tissue (Eq. 2) separately as

follows:

$$\text{NBMF} = \frac{\text{Number of particles in an organism (particles/mg w.w)}}{\text{Number of particles in its prey (particles/mg w.w)}} \quad (1)$$

$$\text{NBMF} = \frac{\text{Number of particles in a fish tissue (particles/mg w.w)}}{\text{Number of particles in the daphnia (particles/mg w.w)}} \quad (2)$$

**Reporting summary**. Further information on research design is available in the Nature Research Reporting Summary linked to this article.

## Data availability
The authors declare that the data supporting the findings of this study are available within the article and its Supplementary Information files. Data are also available from the corresponding authors upon reasonable request.

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

## Acknowledgements

This study was supported by the H2020-MSCA-IF project BTBnano (grant agreement number 793936). The spICP-MS and scICP-MS work was supported by funding from the European Union Horizon 2020 Programme (H2020) under grant agreement number 720952 (project ACEnano). Additional support came from the Ministry of Education, Youth and Sports of the Czech Republic through the CENAKVA project (LM2018099) and the project Sustainable production of healthy fish in various aquaculture systems, PROFISH (CZ.02.1.01/0.0/0.0/16_019/0000869).

## Author contributions

F.A.M. conceptualized, supervised, wrote, and reviewed the study. F.A.M. and L.C. performed the experiment of NM trophic transfer and organisms handling. F.A.M. and D.A.L. performed the experiment of NMs transformation in fish plasma. L.C. extracted plasma from fish blood and contributed to editing the paper. F.A.M. and G.K.D. performed the statistical analysis. F.A.M. and D.A.L. performed TEM imaging. I.L. supervised the biotransformation of NMs in fish plasma. Z.G., P.Z., and G.K.D. contributed to editing the paper. P.M.v.B. contributed to editing the paper. W.J.G.M.P., M.G.V., I.L., and E.V.J. contributed to conceptualizing, supervising, and editing the paper.

## Competing interests

The authors declare no competing interests.

**Additional information**

