## [Peer Review File · Nature Communications]

REVIEWER COMMENTS

Reviewer #1 (Remarks to the Author):

The study was very interesting and the methodology appears to be sound. This reviewer was pleased to find that the authors conducted a TEM analysis of the materials to confirm the size and morphology of the materials.

The observation that transformation of the particles by the daphnia is interesting. It also was novel to observe that ions are likely to be produced in the daphnia and agglomerate into larger particles. Adding additional references and slightly more explanation to support this assertion.

Line 237 – further explanation is needed for the hypothesis for the ions not appearing in the fish. This is a central finding of the paper and it is not adequately supported. Is there a difference in the presence of proteins or other factors that lead to none absorption in fish as compared to the daphnia.

Lines ~245-253 does provide some explanation and addresses previous comment. More information on this method is needed.

The authors suggest no difference in the sizes of particles in the Daphnia compared to fish. What biological significance does this have?

Line 287 should there be a reference to the statistical methods and software used to determine statistical significance.

Figure 3 may be helpful to include mass-based calculations, if possible. Discussion of the mass transfer from daphnia to fish is important in addition to number since regulatory limits on uptake and content of fish and other aquatic organisms are typically based on gravimetric measurements.

Line 346 the authors suggest that the particles pass the blood-brain barrier. This is a significant finding and more detail regarding the certainty and implications of this finding should be discussed.

Overall the study presents key findings regarding the size and form of the Au nanoparticles as they are transferred across the food chain. This reviewer would like to see more discussion on the biological significance of these findings. If this is to be published in a high impact journal, the authors should better explain the importance of these findings in the understanding of environmental toxicology.

Similarly, the findings of the Au nanoparticles migrating to the brain is of considerable significance and more explanation on this finding is needed.

Reviewer #2 (Remarks to the Author):

The authors conducted a study where trophic transfer of total and nano particulate Au was tracked through an artificial laboratory food chain. The authors used sp-ICP-MS as an attempt to differentiate between particulate and ionic Au. All aspects of this research have been reported, and in some cases up to a decade ago. This includes using particle number as a dose metric for trophic transfer. The use of sp-ICP-MS is probably more robust than the previous approaches, but there are also lots of studies that have used sp-ICP-MS to differentiate accumulated NPs in organisms. I think this manuscript contains high quality research, but research that is fairly incremental at this point in time. The manuscript doesn't meet the level of novelty I would expect for nature communications. I think it would be a good contribution to a more subject-oriented journal however.

I am a little bit skeptical of the use of standards in DI water to calibrate for TMAH tissue extracts. Au ions are not very stable in DI water and you typically get a signal suppression relative to standards in HCl or with another stabilizing ligand. This would lead to overestimation of the concentrations.

I'd be a little careful about concluding that particles crossed the blood brain barrier. A considerable

amount of particles can actually be in the vascular compartment of the brain. Knowing the blood concentration and the blood volume in the brain would give you an estimate of how much Au in the whole brain is actually in the vasculature.

Reviewer #3 (Remarks to the Author):

The manuscript examines the bioavailability, biotransformation, and trophic transfer of gold nanoparticles of different size (10, 60, 100 nm) and shape (spherical and rod shaped ((10 × 45 nm and 50 × 100 nm) in a three trophic level aquatic food chain using a novel single-particle (sp) and single-cell (sc) inductively coupled plasma mass spectrometry (ICP-MS). The research team was clearly interdisciplinary, combining expertise in nanoscience, ecotoxicology, and advanced analytical techniques, which would be required to develop the fit-for-purpose workflow needed to conduct this study.

The results of the accumulation of particles of smaller size in algae was consistent with previous studies, while the observation of dissolution and reprecipitation of smaller particles into larger particles in daphnids and the test of the hypothesis for this mechanism was interesting. The observation that a corona of biomolecules on particles accumulated in the third trophic level (fish) verified by spICP-MS to differentiate between Au ions and Au-NMs and the test of this hypothesis was also important new information. The overall contribution of being able to demonstrate the importance of physicochemical properties of internalized NM vs. mass only was novel.

There are a number of areas where the manuscript could be strengthened. There is no mention of other studies employing synchrotron-based x-ray microanalysis for providing other independent information on chemical transformations. The concept of a corona of biomolecules is raised in the case of inhibiting dissolution but not relative to bioavailability or trophic transfer. There are studies in the literature that clearly demonstrate the importance of modification of particles via trophic transfer for enhancing bioavailability and, by extension the potential enhancement of passage through the blood brain barrier. The study by Unrine et al 2012 (Environmental science & technology 46 (17), 9753-9760) and two papers by Judy et al.2011 (Environmental Science & Technology 45 (2), 776-781) and Judy et al., 2012 (Environmental science & technology 46 (22), 12672-12678) demonstrate greater transfer via trophic exposure vs. direct exposure in the presence of the same food. This could be a likely explanation for enhanced uptake in the brain of fish.

The authors thank the reviewers for their careful reading of the manuscript and the insightful comments provided that have helped us to improve the clarity of the manuscript. We provide a point-by-point response to the comments below.

REVIEWER COMMENTS

Reviewer #1:

The study was very interesting and the methodology appears to be sound. This reviewer was pleased to find that the authors conducted a TEM analysis of the materials to confirm the size and morphology of the materials. The observation that transformation of the particles by the daphnia is interesting.

We thank the reviewer for his/her positive comments on our manuscript.

1. It also was novel to observe that ions are likely to be produced in the daphnia and agglomerate into larger particles. Adding additional references and slightly more explanation to support this assertion.

We thank the reviewer for his/her suggestion. We have added some more references and more explanations of the results and particularly highlighted the biological significance of our findings (Line 208-209, 223-226).

“For example, Briffa et al. showed that the dissolution of cerium dioxides (CeO₂)-NMs in the exposure medium of daphnids is negligible, whereas at simulated gut conditions of daphnids, there is around 40% Ce dissolution from the particles.”

“The dissolution of the Au-NMs in the daphnids could have occurred in the gut, which has a lower pH (~ 4) than the exposure medium. Absorption of a biomolecule corona to the surface of NMs when they enter the daphnids can also accelerate the dissolution of the particles if there is a strongly Au-binding protein present, or can slow dissolution by promoting sulfidation processes.”

2. Line 237 – further explanation is needed for the hypothesis for the ions not appearing in the fish. This is a central finding of the paper and it is not adequately supported. Is there a difference in the presence of proteins or other factors that lead to none absorption in fish as compared to the daphnia? Lines ~245-253 does provide some explanation and addresses previous comment. More information on this method in needed.

We thank the reviewer for this suggestion. More explanation now is given for why the Au ions did not appear to bioaccumulate in fish (Line 259-263, 266-269).

“Two explanations can be put forward for these findings. First, the Au ions in daphnids may not be bioavailable to fish and consequently they were not taken up by fish. Many studies have already shown that the distribution of soluble metals in organisms influences their trophic transfer by controlling their bioavailability for predators. For example, soluble metals distributed in subcellular organelles are more efficiently taken up by predators than are metals

present in insoluble fractions within prey, such as in metal-rich granules. It is likely that the ionic form of the Au in daphnids was stored in fractions that make them non-bioavailable to fish. Second, the absence of Au ions released from the particles in fish could be due to the evolution of the biomolecule corona on the surface of the particles when entering the fish body. The corona may reduce the Au release from Au-NMs in fish unlike in daphnids, as observed for other NMs in other organisms.”

3. The authors suggest no difference in the sizes of particles in the Daphnia compared to fish. What biological significance does this have?

We found that the size of some of the particles in the daphnia was similar to those in fish. This finding indicates that some of the Au-NMs may not undergo a further transformation in fish, suggesting that the transformation of metallic NMs in organisms could be a species-dependent phenomenon. For instance, in daphnids, some of the particles dissolved and agglomerated, while they did not transform in fish. This finding raises questions regarding the application of model organisms to assess the risk of NMs as each organism may have different influences on the biotransformation of NMs and the findings may not be extrapolable to other organisms. We have added this explanation to the text to show the biological significance of the finding (Lines 285-290).

4. Line 287 should there be a reference to the statistical methods and software used to determine statistical significance.

We thank the reviewer for his/her suggestion. We have added a reference for the statistical methodology (Line 600)

5. Figure 3 may be helpful to include mass-based calculations, if possible. Discussion of the mass transfer from daphnia to fish is important in addition to number since regulatory limits on uptake and content of fish and other aquatic organisms are typically based on gravimetric measurements.

We totally agree with the reviewer that the mass-based data are important. All these data are presented in Table 1 of the manuscript, Table S4 of the Supporting Information and Figure S5 of the Supporting Information.

6. Line 346 the authors suggest that the particles pass the blood-brain barrier. This is a significant finding and more detail regarding the certainty and implications of this finding should be discussed.

We discuss this finding in more details and explain the uncertainty and implications of our findings with regard to Au-NMs in the fish brain (Line 379-387). The potential for the NMs to be contained within the endothelial cell layer of the blood-brain barrier is also noted.

“It is possible that some of the NMs could be in the vascular compartment of the brain, i.e. the endothelial cells that comprise the blood-brain barrier itself, which has been demonstrated in vitro to be a site of NM accumulation 56. Because the workflow presented in this study has been developed using a set of robust techniques to distinguish between particle and ions, we were able to reproducibly measure the number of Au-NMs in the fish brain tissue (including the endothelial barrier). In general, understanding the number of NMs in the brain is important

to determine the effective dose because NMs have a larger surface area than the corresponding bulk material, which means a higher number of molecules on the surface of NM are available for interaction with surrounding biological material”

7. Overall the study presents key findings regarding the size and form of the Au nanoparticles as they are transferred across the food chain. This reviewer would like to see more discussion on the biological significance of these findings. If this is to be published in a high impact journal, the authors should better explain the importance of these findings in the understanding of environmental toxicology.

We thank the reviewer for his/her suggestion and comments which helped us to improve the manuscript. We have added some more explanation with regard to the biological significance of the findings (Line 236-239, Line 255-257, 349-351).

“This biotransformation of metallic NMs in organisms is of paramount importance for understanding the toxicity of NMs as it determines the behaviour and biological fate of the NMs and the release of ions from the particles and, subsequently, the target organs reached by the different metal forms”

“By modifying the size of the particles, daphnids can dramatically influence the biological fate of NMs in the higher trophic levels and how metallic NMs distribute over the entire aquatic food chain.”

“This indicates that only a small portion of the spherical 10 nm and rod-shaped 10 × 45 nm transfer to fish compared to the other Au-NMs tested. This suggests that particle size plays a significant role in the trophic transfer of NMs and the bioavailability of small NMs to fish may be lower than the bioavailability of their larger counterparts.”

8. Similarly, the findings of the Au nanoparticles migrating to the brain is of considerable significance and more explanation on this finding is needed.

We have added some more explanation to highlight the hazard associated with migration of nanoparticles to the brain (Line 379-387).

“It is possible that some of the NMs could be in the vascular compartment of the brain, i.e. the endothelial cells that comprise the blood-brain barrier itself, which has been demonstrated in vitro to be a site of NM accumulation. Because the workflow presented in this study has been developed using a set of robust techniques to distinguish between particle and ions, we were able to reproducibly measure the number of Au-NMs in the fish brain tissue (including the endothelial barrier). In general, understanding the number of NMs in the brain is important to determine the effective dose because NMs have a larger surface area than the corresponding bulk material, which means a higher number of molecules on the surface of NM are available for interaction with surrounding biological material”

Reviewer #2:

The authors conducted a study where trophic transfer of total and nano particulate Au was tracked through an artificial laboratory food chain. The authors used sp-ICP-MS as an attempt to differentiate between particulate and ionic Au. All aspects of this research have been reported, and in some cases up to a decade ago. This includes using particle number as a dose metric for trophic transfer. The use of sp-ICP-MS is probably more robust than the previous approaches, but there are also lots of studies that have used sp-ICP-MS to differentiate accumulated NPs in organisms. I think this manuscript contains high quality research, but research that is fairly incremental at this point in time. The manuscript doesn't meet the level of novelty I would expect for nature communications. I think it would be a good contribution to a more subject-oriented journal however.

We thank the reviewer for his/her positive comments on our manuscript. We must emphasize that this manuscript used an interdisciplinary approach, as was highlighted by the third reviewer, to tackle some of the challenges in the field to provide new insight into the trophic transfer of NMs by considering the biotransformation of NM in each trophic level. The study provides information on how physicochemical properties of NMs influence their biotransformation, thus, trophic transfer. By considering particle number as a dose metric, the manuscript opens new horizons not only for understanding the biological fate of NMs as a function of their properties but also for the environmental risk assessment of NMs where the particle number is an important element.

1. I am a little bit skeptical of the use of standards in DI water to calibrate for TMAH tissue extracts. Au ions are not very stable in DI water and you typically get a signal suppression relative to standards in HCl or with another stabilizing ligand. This would lead to overestimation of the concentrations.

We agree with the reviewer and we appreciate his/her concern. In this study, we have applied Milli-Q water to provide standards for calibration of the ICP-MS for measuring the total mass of Au in the samples. The standards were provided on a daily basis (containing 1% acid nitric) and used immediately to calibrate the machine. For the TMAH, we did not use Au ions, as the purpose of applying TMAH was to extract the nanoparticles from the tissues and not the Au ions. We have added the calibration curve to the Supporting Information (Figure S7).

2. I'd be a little careful about concluding that particles crossed the blood brain barrier. A considerable amount of particles can actually be in the vascular compartment of the brain. Knowing the blood concentration and the blood volume in the brain would give you an estimate of how much Au in the whole brain is actually in the vasculature.

We fully agree with the reviewer and have modified the discussion at lines 379-387, to indicate that the particles may in fact be accumulating in the endothelial cells of the blood vessels constituting the blood brain barrier.

Reviewer #3:

The research team was clearly interdisciplinary, combining expertise in nanoscience, ecotoxicology, and advanced analytical techniques, which would be required to develop the fit-for-purpose workflow needed to conduct this study. The results of the accumulation of particles of smaller size in algae was consistent with previous studies, while the observation of dissolution and reprecipitation of smaller particles into larger particles in daphnids and the test of the hypothesis for this mechanism was interesting. The observation that a corona of biomolecules on particles accumulated in the third trophic level (fish) verified by spICP-MS to differentiate between Au ions and Au-NMs and the test of this hypothesis was also important new information. The overall contribution of being able to demonstrate the importance of physicochemical properties of internalized NM vs. mass only was novel.

We thank the reviewer for his/her positive comments on this study.

1. There is no mention of other studies employing synchrotron-based x-ray microanalysis for providing other independent information on chemical transformations.
Thank you for the suggestion. We have added some information, acknowledging that techniques are emerging to explain the metallic speciation of nanoparticles within tissues. We have also reviewed the technique in one of our last publications (<https://doi.org/10.1016/j.scitotenv.2019.01.105>), indicating that we are aware that synchrotron-based X-ray analysis is used already for quite a while. However, synchrotron-based X-ray analysis has important limitations for nanoparticle applications in organisms, because of its high detection limit and the lack of ability to provide information on particle size. We have added this explanation as lines 210-217 of the text.
2. The concept of a corona of biomolecules is raised in the case of inhibiting dissolution but not relative to bioavailability or trophic transfer.
Indeed, proteins can change the behavior, biotransformation and biological fate (bioavailability, bioaccumulation, biodistribution, clearance, etc.) of nanoparticles in organisms. However, in this study the aim was to understand the trophic transfer of nanoparticles from algae to fish. The algae were the first organisms in this chain. It means that algae are exposed to pristine nanoparticles brought into the exposure medium. The algae therefore are not exposed to biological corona-coated nanoparticles (biological corona is acquired when nanoparticles enter physiological media of organisms). We could not directly expose fish to corona-coated nanoparticles to study the bioavailability because the bioavailability resulting from exposure through the food chain is dramatically different from bioavailability as a result of water born exposure. Therefore, we were not able to discuss corona in the context of bioavailability or trophic transfer. However, we could investigate the dissolution because it was possible to simulate the physiological medium of fish and no exposure was required.
3. There are studies in the literature that clearly demonstrate the importance of modification of particles via trophic transfer for enhancing bioavailability and, by extension the potential enhancement of passage through the blood brain barrier. The study by Unrine et al 2012 (Environmental science & technology 46 (17), 9753-9760) and two papers by Judy et al. 2011 (Environmental Science & Technology 45 (2), 776-781) and Judy et al., 2012 (Environmental science & technology 46 (22), 12672-12678) demonstrate greater transfer via trophic exposure vs. direct exposure in the presence of the same food. This could be a likely explanation for enhanced uptake in the brain of fish.
We thank the reviewer for suggesting some studies to be used as references in the current manuscript, which improves the interpretation of the results significantly. We have added the studies in lines 172-174, 393-395.

REVIEWER COMMENTS

The revision was seen by the relevant reviewers who did not provide any further comments.